# ANALYZING THE ROLE OF MODEL UNCERTAINTY FOR ELECTRONIC HEALTH RECORDS

## ABSTRACT

In medicine, both ethical and monetary costs of incorrect predictions can be significant, and the complexity of the problems often necessitates increasingly complex models. Recent work has shown that changing just the random seed is enough for otherwise well-tuned deep neural networks to vary in their individual predicted probabilities. In light of this, we investigate the role of model uncertainty methods in the medical domain. Using recurrent neural network (RNN) ensembles and various Bayesian RNNs, we show that population-level metrics, such as AUC-PR, AUC-ROC, log-likelihood, and calibration error, do not capture model uncertainty. Meanwhile, the presence of significant variability in patient-specific predictions and optimal decisions motivates the need for capturing model uncertainty. Understanding the uncertainty for individual patients is an area with clear clinical impact, such as determining when a model decision is likely to be brittle. We further show that RNNs with only Bayesian embeddings can be a more efficient way to capture model uncertainty compared to ensembles, and we analyze how model uncertainty is impacted across individual input features and patient subgroups.

## 1 INTRODUCTION

Machine learning has found great and increasing levels of success in the last several years on many well-known benchmark datasets. This has led to a mounting interest in non-traditional problems and domains, each of which bring their own requirements. In medicine specifically, individualized predictions are of great importance to the field (Council et al., 2011), and there can be severe costs for incorrect predictions and decisions due to the risk to human life and the associated ethical concerns (Gillon, 1994).

Existing state-of-the-art approaches using deep neural networks in medicine often make use of either a single model or an average over a small ensemble of models, focusing on improving the accuracy of probabilistic predictions (Harutyunyan et al., 2017; Rajkomar et al., 2018b; Xu et al., 2018; Choi et al., 2018). These works, while focusing on capturing the data uncertainty, do not address the *model* uncertainty that is inherent in fitting deep neural networks. For example, when predicting patient mortality in an ICU setting, existing approaches might be able to achieve high AUC-ROC, but will be unable to differentiate between patients for whom the model is *certain* about its probabilistic prediction, and those for whom the model is fairly *uncertain*.

In this paper, we examine the use of model uncertainty specifically in the context of predictive medicine. Model uncertainty has made many methodological advances in recent years—including reparameterization-based variational Bayesian neural networks (Blundell et al., 2015; Kucukelbir et al., 2017; Louizos & Welling, 2017), Monte Carlo dropout (Gal & Ghahramani, 2016), ensembles (Lakshminarayanan et al., 2017), and function priors (Hafner et al., 2018; Garnelo et al., 2018; Malinin & Gales, 2018). Deep neural networks combined with advanced model uncertainty methods can directly impact clinical care by answering several questions that naturally occur in predictive medicine:

- How do the realized functions in any of the approaches, such as individual models in the ensemble approach, compare in terms of population-level metric performance such as AUC-PR, AUC-ROC, or log-likelihood?
- If and how does model uncertainty assist in calibrating predictions?

- How does model uncertainty change across different patient subgroups, in terms of ethnicity, gender, age, or length of stay?
- How do various feature values contribute towards model uncertainty?
- How does model uncertainty affect optimal decisions made under a given clinically-relevant cost function?

**Contributions**   Using sequence models on the MIMIC-III clinical dataset (Johnson et al., 2016), we make several important findings. For the ensembling approach of quantifying model uncertainty, we find that the models within the ensemble can collectively exhibit a wide variability in predicted probabilities for some patients, despite being well-calibrated and having *nearly identical dataset-level metric performance*. We find that this even extends into the space of optimal decisions. That is, models with nearly equivalent metric performance can disagree significantly on the final decision, thus transforming an "optimal" decision into a random variable. Significant variability in patient-specific predictions and decisions can be an indicator of when a model decision is likely to be brittle, and we show that using a single model or an average over models can mask this information. This motivates the importance of model uncertainty for clinical decision systems. Given this, we proceed with an analysis over different clinical tasks and datasets, looking at how model uncertainty is impacted across individual input features and patient subgroups. We then show that models with Bayesian embeddings can be a more efficient way to capture model uncertainty compared to deep ensembles.

## 2   BACKGROUND

**Data uncertainty**   Data uncertainty can be viewed as uncertainty regarding a given outcome due to incomplete information, and is also known as "output uncertainty" or "risk" (Knight, 1957). For binary tasks, this equates to a single probability value. More specifically, this can be described as

$$y \sim \text{Bernoulli}(\lambda), \quad \lambda = f(\mathbf{x}, \mathbf{w}), \tag{1}$$

where the model $f$, as a function of the inputs $\mathbf{x}$ and parameters $\mathbf{w}$, outputs the parameter $\lambda$ for a Bernoulli distribution representing the conditional distribution $p(y|\mathbf{x}, \mathbf{w})$ for the outcome $y$.

**Model uncertainty**   Model uncertainty can be viewed as uncertainty in the correct values of the parameters for the predictive outcome distribution due to lack of knowledge of the true function. For binary tasks, this equates to a distribution of plausible probability values for a Bernoulli distribution, corresponding to a set of plausible functions. More specifically, this can be described as

$$y \sim \text{Bernoulli}(\lambda), \quad \lambda \sim p(\lambda|\mathbf{x}, \mathbf{w}), \tag{2}$$

where there is an induced distribution over the Bernoulli parameter $\lambda$ for a given example that represents uncertainty in the true outcome distribution due to uncertainty in function space. For the remainder of the paper, we will use the phrase *predictive uncertainty distribution* to refer to this distribution over the parameter of the outcome distribution.

**Calibration**   A model is said to be perfectly calibrated if, for all examples for which the model produces the same prediction $p$ for some outcome, the percentage of those examples truly associated with the outcome is equal to $p$, across all values of $p$. If a model is systematically over- or under-confident, it can be difficult to reliably use its predicted probabilities for decision making. The expected calibration error (ECE) metric (Naeini et al., 2015) is one tractable way to approximate the calibration of a model given a finite dataset.

**Deep Ensembles**   Deep ensembles (Lakshminarayanan et al., 2017) is a method for quantifying model uncertainty. In this approach, an ensemble of $M$ deterministic[1] neural networks (NNs) is trained by varying only the random seed of an otherwise well-tuned set of hyperparameters. Given this ensemble, predictions can be made with each model $m$ for a given example $i$, where (for a binary task) each prediction is the probability parameter $\lambda_m^{(i)}$ for the Bernoulli distribution over the outcome. The set of $M$ probabilistic predictions $\{\lambda_1^{(i)}, \lambda_2^{(i)}, \ldots, \lambda_M^{(i)}\}$ for the same example can then be viewed as the distribution $p(\lambda|\mathbf{x}, \mathbf{w})$ over $\lambda^{(i)}$, where this distribution represents model uncertainty.

---

[1]We use the term "deterministic" to refer to the usual setup in which we optimize the parameter values of our function directly, yielding a trained model with fixed parameter values at test time.

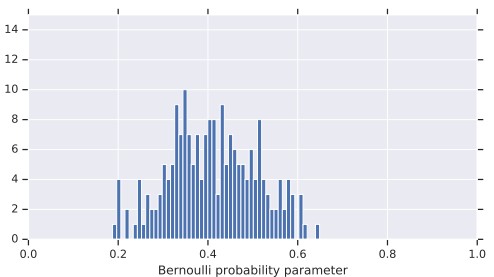

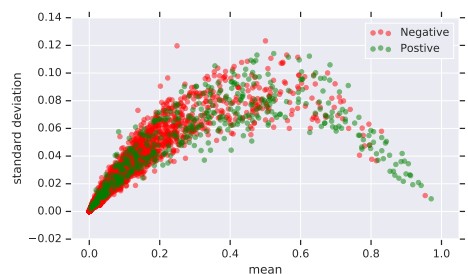

**Figure 1:** A histogram of predictions from $M$ deterministic RNN models trained with different random seeds for a given ICU patient's mortality probability. As shown here, model uncertainty can cause a high-variance predictive distribution across individual models in an ensemble. This is not captured when using a single model or an average over an ensemble.

**Figure 2:** A plot of the mean versus standard deviation of the predictive uncertainty distributions of the deterministic ensemble for positive and negative patients in the validation set. We find that the standard deviations do not form a simple linear relationship with the mean. For reference, we note that the variances of the distributions are generally lower than that of a Bernoulli distribution's variance curve.

**Bayesian RNNs**   Bayesian recurrent neural networks (RNNs) (Fortunato et al., 2017) are RNNs with a prior distribution placed over the parameters. This allows us to express model uncertainty as uncertainty over the true values for the parameters in the model, i.e., "weight uncertainty" (Blundell et al., 2015). By introducing a distribution over all, or a subset, of the weights in the model, we can induce different functions, and thus different outcomes, through realizations of different weight values via draws from the posterior distributions. This allows us to empirically capture model uncertainty in the predictive uncertainty distribution $p(\lambda|\mathbf{x}, \mathbf{w})$ by drawing $M$ samples from a single Bayesian RNN for a given example.

## 3   MEDICAL UNCERTAINTY

### 3.1   CLINICAL TASKS

We first show results using Medical Information Mart for Intensive Care (MIMIC-III) (Johnson et al., 2016), a publicly available EHR dataset, in terms of two tasks: in-patient mortality prediction, and multiclass diagnosis prediction at discharge, where diagnosis codes are represented by the single-level Clinical Classifications Software (CCS) codes. Similar to Rajkomar et al. (2018b), we train deep RNNs on MIMIC-III, which is collected from 46,520 patients admitted to intensive care units (ICUs) at Beth Israel Deaconess Medical Center, where 9,974 expired during the encounter (*i.e.,* 1:4 ratio between positive and negative samples). In order to demonstrate that our findings generalize, we additionally experiment with eICU Collaborative Research Database (eICU) (Pollard et al., 2018), another publicly available EHR dataset, in terms of in-patient mortality prediction. Our model embeds and aggregates a patient's time-series features (*e.g.* medications, lab measures) and global features (*e.g.* gender, age), feeds them to one or more long short-term memory (LSTM) layers (Schmidhuber & Hochreiter, 1997), followed by hidden and output affine layers. See the Appendix for more details.

Existing deep learning approaches in predictive medicine focus on capturing data uncertainty, namely accurately predicting the risk $\lambda = f(\mathbf{x}, \mathbf{w})$ of a patient's mortality (*i.e.,* how likely is the patient to expire?). This work, on the other hand, also focuses on addressing the model uncertainty aspect of deep learning, namely the distribution $\lambda \sim p(\lambda|\mathbf{x}, \mathbf{w})$ over the risk of mortality for a patient (*i.e.,* are there alternative risk predictions, and if so, how diverse are they?).

### 3.2   CHOICE OF UNCERTAINTY METHODS

To quantify model uncertainty for clinical tasks, we explore the use of deep RNN ensembles and various Bayesian RNNs. For the deep ensembles approach, we optimize for the ideal hyperparameter values for our RNN model via black-box Bayesian optimization (Golovin et al., 2017), and then

train $M$ replicas of the best model. Only the random seed differs between the replicas. At prediction time, we make predictions with all $M$ models for each patient. For the Bayesian RNNs, we train a single model, and then draw $M$ samples from it at prediction time. To train the Bayesian RNN, we take a variational inference approach by adapting our RNNs to use factorized weight posteriors $q(\mathbf{w}|\boldsymbol{\theta}) = \prod_i q(\mathbf{w}_i|\boldsymbol{\theta}_i)$, where weight tensors $\mathbf{w}_i$ in the models are represented by normal distributions with learnable mean and diagonal covariance parameters represented as $\boldsymbol{\theta}_i$. Normal distributions with zero mean and tunable standard deviation are used as weight priors. We train our models by minimizing the Kullback-Leibler (KL) divergence

$$
\begin{aligned}
\mathcal{L}(\boldsymbol{\theta}) &= \mathrm{KL}[q(\mathbf{w}|\boldsymbol{\theta}) \,\|\, p(\mathbf{w}|\mathbf{y}, \mathbf{X})] \\
&= \mathrm{KL}[q(\mathbf{w}|\boldsymbol{\theta}) \,\|\, p(\mathbf{w})] - \mathbb{E}_{q(\mathbf{w}|\boldsymbol{\theta})}\left[\ln p(\mathbf{y}|\mathbf{X}, \mathbf{w})\right]
\end{aligned}
\tag{3}
$$

between the approximate weight posterior and the true, but unknown posterior, which overall equates to minimizing an expectation over the usual negative log likelihood term plus a KL regularization term. To easily shift between the deterministic and Bayesian models, we make use of the Bayesian Layers (Tran et al., 2018) abstractions.

### 3.3 Optimal Decisions via Sensitivity Requirements

The key desire in clinical practice is to make a decision based on the model's predicted probability and its associated uncertainty. Given a set of potential outcomes $y_k$, a set of conditional probabilities $p(y_k|\mathbf{x})$ for the given outcomes, and the associated costs $L_{kj}$ for either correctly or incorrectly predicting the outcome, an optimal decision can be determined by minimizing the expected decision loss

$$
\mathbb{E}[L] = \sum_k \sum_j \int_{\mathcal{R}_j} L_{kj} p(y_k|\mathbf{x}) p(\mathbf{x}) \, \mathrm{d}\mathbf{x},
\tag{4}
$$

where $\mathcal{R}_j$ is the decision region for assigning example $\mathbf{x}$ to class $j$, and $p(\mathbf{x})$ is the density of $\mathbf{x}$ (Bishop, 2006).

Designing elaborate decision cost functions for clinical applications is not an easy task as it requires expert knowledge on the prediction target, cost-benefit analysis, and medical resource allocation. Fortunately we can use a clinically relevant alternative, which is the *sensitivity requirement*. Often in clinical research, certain sensitivity (*i.e.,* recall) levels must be met when making predictions in order for a model to be clinically relevant. The goal in such cases is to maximize the precision while still maintaining the required sensitivity level. Viewed as a decision cost function, the cost is infinite if the recall is below the target level, and is otherwise minimized as the precision is increased, where the optimized parameter is the global probability threshold.

For each of the $M$ models in our ensemble, we can optimize the sensitivity-based decision cost function and make optimal decisions for all examples. Thus, for each example, there will be a set of $M$ optimal decisions, forming a distribution. The optimal decision $d$ then becomes a random variable

$$
d \sim \mathrm{Bernoulli}(\phi), \quad \phi = \frac{1}{M} \sum_{\lambda \in \Lambda} \mathbb{1}(\lambda_j \geq t)),
\tag{5}
$$

where $\phi$ is the percentage of the set $\Lambda = \{\lambda_1, \lambda_2, \ldots, \lambda_M\}$ of $M$ probability values for a given example that are greater than or equal to the optimized decision threshold $t$.

## 4 Experiments

We perform four sets of experiments. First, in order to demonstrate the importance of quantifying uncertainty in predictive medicine, we examine individual models in the RNN ensemble in terms of prediction metrics, calibration, uncertainty distributions, and decision-making. Second, we examine multiple variants of Bayesian RNNs to understand where uncertainty in the model matters most, comparing them with the deterministic ensemble counterpart. Third, we use the deterministic RNN ensemble to examine uncertainty across different patient subgroups. Finally, we analyze the Bayesian RNN with embedding distributions to examine uncertainty across individual features.

**Table 1:** Dataset-level metrics for the MIMIC-III binary mortality and multiclass CCS prediction tasks across $M = 200$ models in the deterministic RNN ensemble. Individual models are nearly identical in terms of dataset-level performance across both tasks, but selecting a single model would remove the model uncertainty information such as that visualized in Figure 1.

| TASK | METRIC | VALIDATION | TEST |
|------|--------|------------|------|
| MORTALITY | AUC-PR | 0.4496 (0.0025) | 0.3886 (0.0059) |
| | AUC-ROC | 0.8753 (0.0019) | 0.8623 (0.0031) |
| | ECE | 0.0176 (0.0040) | 0.0162 (0.0043) |
| | ACE | 0.0210 (0.0042) | 0.0233 (0.0057) |
| CCS | TOP-5 RECALL | 0.7126 (0.0071) | 0.7090 (0.0088) |
| | TOP-5 PRECISION | 0.1425 (0.0014) | 0.1418 (0.0018) |
| | TOP-5 F1 | 0.2375 (0.0024) | 0.2363 (0.0029) |
| | LOG-LIKELIHOOD | -5.1040 (0.0075) | -5.1081 (0.0083) |
| | ECE | 0.0446 (0.0072) | 0.0499 (0.0082) |
| | ACE | 4.2189E-3 (7.3111E-8) | 4.2191E-3 (7.6136E-8) |

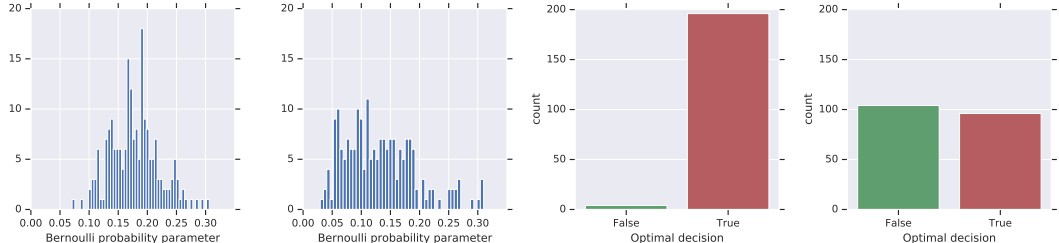

**Figure 3: Left two:** The deterministic ensemble produces different mortality prediction distributions (*i.e.* model uncertainty) for two different patients from the validation set. **Right two:** The corresponding optimal decision distributions, "True" being the positive prediction for mortality and "False" being the opposite. For some patients the ensemble is relatively certain about the optimal decision, while for other patients there is a large amount of uncertainty.

## 4.1 WHEN DO WE OBSERVE UNCERTAINTY?

**Clinical Metrics** For our clinical tasks, we first measure the dataset-level metrics: area under the precision-recall curve (AUC-PR), area under the receiver operating characteristic curve (AUC-ROC), top-5 recall, top-5 precision, top-5 F1, held-out log-likelihood, ECE (Naeini et al., 2015), and adaptive calibration error (ACE) (Nixon et al., 2019). Table 1 shows the performance averaged over individual models in our deterministic ensemble, with standard deviation in the parenthesis. Interestingly, individual models are overall well-calibrated and nearly equivalent in terms of performance. If we were to choose only one model in practice based on the dataset-level metrics, it is highly likely any one could be selected as a good model, and we would lose the model uncertainty information.

**Predictive Uncertainty Distributions & Statistics** Knowing that the models in our ensemble are well-calibrated and effectively equivalent in terms of performance, we turn to making predictions for individual examples. Figure 1 visualizes the predictive uncertainty distribution for a single patient. We find that there is a wide variability in predicted Bernoulli probabilities for some patients (with spreads as high as $57.5\%$). Ignoring this variance through the use of either a single deterministic model or an average over an ensemble is likely harmful since it prevents this uncertainty from being conveyed to a physician. Figure 2 visualizes the means versus standard deviations of the predictive uncertainty distributions for the deterministic ensemble on all validation set examples. In contrast to the variance of a Bernoulli distribution, which is a simple function of the mean, we find that the standard deviations are patient-specific, and thus cannot be determined a priori.

**Optimal Decision Distributions & Statistics** In practice, model uncertainty is important as it can affect one's decisions. To test this, we optimize the sensitivity (*i.e.* recall) based decision cost function with respect to the probability threshold for each model in our RNN ensemble separately

**Table 2:** Metrics for marginalized predictions on the MIMIC-III and eICU mortality tasks given $M = 200$ models in the deterministic RNN ensemble, and $M = 200$ samples from each of the Bayesian RNN models. Confidence intervals are computed via validation and test set bootstrapping with 1000 bootstrap sets.

| Data | Model | Val. AUC-PR | Val. AUC-ROC | Val. NLL | Test AUC-PR | Test AUC-ROC | Test NLL |
|---|---|---|---|---|---|---|---|
| MIMIC-III | Deterministic Ensemble | 0.4564 ($\pm$1e−3) | 0.8774 ($\pm$5e−4) | 0.7113 ($\pm$7e−5) | 0.3921 ($\pm$1e−3) | 0.8643 ($\pm$5e−4) | 0.7148 ($\pm$7e−5) |
| | Bayesian Embeddings | 0.4580 ($\pm$1e−3) | 0.8776 ($\pm$4e−4) | 0.7152 ($\pm$7e−5) | 0.3977 ($\pm$2e−3) | 0.8612 ($\pm$5e−4) | 0.7186 ($\pm$7e−5) |
| | Bayesian Output | 0.4382 ($\pm$2e−3) | 0.8714 ($\pm$5e−4) | 0.7058 ($\pm$7e−5) | 0.3702 ($\pm$1e−3) | 0.8572 ($\pm$5e−4) | 0.7087 ($\pm$7e−5) |
| | Bayesian Hidden+Output | 0.4492 ($\pm$1e−3) | 0.8751 ($\pm$5e−4) | 0.7118 ($\pm$7e−5) | 0.3893 ($\pm$1e−3) | 0.8607 ($\pm$5e−4) | 0.7149 ($\pm$7e−5) |
| | Bayesian RNN+Hidden+Output | 0.4396 ($\pm$2e−3) | 0.8673 ($\pm$5e−4) | 0.7097 ($\pm$7e−5) | 0.3860 ($\pm$2e−3) | 0.8542 ($\pm$5e−4) | 0.7125 ($\pm$7e−5) |
| | Fully Bayesian | 0.4354 ($\pm$2e−3) | 0.8692 ($\pm$5e−4) | 0.7100 ($\pm$6e−5) | 0.3829 ($\pm$1e−3) | 0.8552 ($\pm$5e−4) | 0.7133 ($\pm$6e−5) |
| eICU | Deterministic Ensemble | 0.1951 ($\pm$1e−3) | 0.7882 ($\pm$7e−4) | 0.1435 ($\pm$3e−4) | 0.2196 ($\pm$1e−3) | 0.7868 ($\pm$6e−4) | 0.2435 ($\pm$5e−4) |
| | Bayesian Embeddings | 0.1996 ($\pm$1e−3) | 0.7807 ($\pm$1e−4) | 0.1455 ($\pm$4e−4) | 0.2244 ($\pm$1e−3) | 0.7733 ($\pm$7e−4) | 0.1620 ($\pm$4e−4) |
| | Bayesian Output | 0.1738 ($\pm$1e−3) | 0.7677 ($\pm$7e−4) | 0.1664 ($\pm$3e−4) | 0.1942 ($\pm$1e−3) | 0.7580 ($\pm$7e−4) | 0.1810 ($\pm$4e−4) |
| | Bayesian Hidden+Output | 0.1712 ($\pm$1e−3) | 0.7801 ($\pm$7e−4) | 0.1619 ($\pm$3e−4) | 0.2140 ($\pm$e−) | 0.7817 ($\pm$6e−4) | 0.1713 ($\pm$3e−4) |
| | Bayesian RNN+Hidden+Output | 0.1675 ($\pm$1e−3) | 0.7791 ($\pm$7e−4) | 0.1477 ($\pm$3e−4) | 0.2147 ($\pm$1e−3) | 0.7809 ($\pm$7e−4) | 0.1583 ($\pm$3e−4) |
| | Fully Bayesian | 0.2004 ($\pm$1e−3) | 0.7910 ($\pm$7e−4) | 0.1377 ($\pm$3e−4) | 0.2280 ($\pm$1e−3) | 0.7818 ($\pm$7e−4) | 0.1541 ($\pm$4e−4) |

to achieve a recall of $70\%$, and then make optimal decisions for each example with each of the $M$ models. Figure 3 visualizes how model uncertainty in probability space is realized in optimal decision space for two patients. We see that the model uncertainty does indeed extend into the optimal decision space, converting the optimal decision into a random variable. Furthermore, the decision distribution's variance can be quite high, and knowing when this is the case is important in order to avoid the cost of any incorrect decisions made by the system due to lack of precise knowledge about the correct level of data uncertainty.

## 4.2 Comparison: Variants of Bayesian RNNs and Deterministic RNN Ensembles

A natural question in practice when employing the Bayesian approach is which part of the model should capture model uncertainty. To do so, we study Bayesian RNNs under a variety of priors:

- **Bayesian Embeddings** A Bayesian RNN in which the embedding parameters are stochastic, and all other parameters are deterministic.

- **Bayesian Output** A Bayesian RNN in which the output layer parameters are stochastic, and all other parameters are deterministic.

- **Bayesian Hidden+Output** A Bayesian RNN in which the hidden and output layer parameters are stochastic, and all other parameters are deterministic.

- **Bayesian RNN+Hidden+Output** A Bayesian RNN in which the LSTM, hidden, and output layer parameters are stochastic, and all other parameters are deterministic.

- **Fully Bayesian** A Bayesian RNN in which all parameters are stochastic.

Table 2 displays the metrics over marginalized predictions for each of the Bayesian RNN models and the deterministic RNN ensemble on the MIMIC-III and eICU mortality tasks. We find that the Bayesian Embeddings RNN model outperforms all other Bayesian variants and slightly outperforms the deterministic ensemble in terms of AUC-PR for MIMIC-III, and that the fully-Bayesian RNN outperforms the other models on the eICU dataset. Additionally, all of the Bayesian variants are either comparable or outperform the deterministic ensemble in terms of held-out log-likelihood on both datasets. Since our Bayesian models achieve strong performance while only requiring training

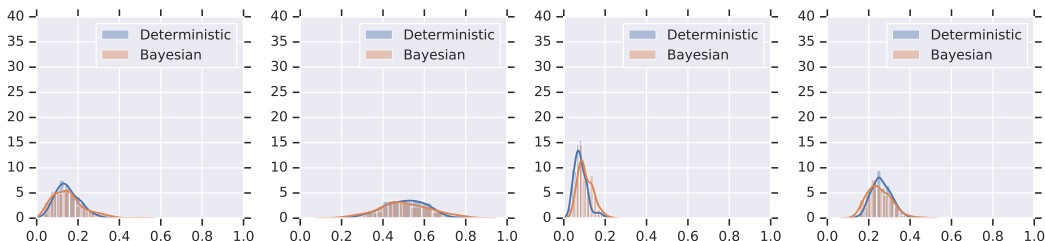

**Figure 4:** Predictive uncertainty distributions of both the RNN with Bayesian embeddings and the deterministic RNN ensemble for individual patients. We find that the Bayesian model is qualitatively able to capture model uncertainty that is quite similar to that of the ensemble with considerably smaller number of parameters
.

of a single model (7.22 million parameters in the MIMIC-III Bayesian Embeddings RNN), versus $M$ models in the deterministic ensemble ($200 \times 6.16$ million parameters), as well as only requiring the single model at prediction time, using Bayesian RNNs can be a more efficient approach.

Figure 4 visualizes the predictive distributions of both the Bayesian RNN and the deterministic RNN ensemble for four individual patients. We find that the Bayesian model is qualitatively able to capture model uncertainty that is quite similar to that of the deterministic ensemble. Overall, the Bayesian Embeddings RNN, compared to the deterministic ensemble, demonstrated slightly better predictive performance and qualitatively similar uncertainty performance, with considerably less computational resources, making it an attractive choice in clinical practice.

### 4.3 PATIENT SUBGROUP ANALYSIS

We next turn to an exploration of the effects of model uncertainty across patient subgroups. We split validation set encounters into subgroups by demographic characteristics, namely patient gender (3089 male vs. 2548 female) or age (adults divided into quartiles of 1216, with a separate fifth group of 773 neonates). For this analysis, we focus on the deterministic RNN ensemble described in Section 4.1, as the Bayesian models sample $M = 200$ weights separately for each prediction rather than globally for the complete validation set. For each model in the ensemble, we compute validation set performance metrics separately over each subgroup and then compute the correlation between these metrics over all models in the ensemble, to evaluate whether the ensemble models tend to specialize to one or more subgroups at the cost of performance on others. We find some evidence of this phenomenon: for example, AUC-PR for male patients is negatively correlated with AUC-PR for female patients (Pearson's $r = -0.442$, see Figure 5), and AUC-PR for the oldest quartile of adult patients is somewhat negatively correlated with AUC-PR for other adults or for neonates (Pearson's $r$ between $-0.18$ and $-0.37$).

We also compare uncertainty metrics across subgroups, including standard deviation and range of the predictive uncertainty distributions and variance of the optimal decision distributions for patients in each subgroup. For this analysis, we examine both the deterministic ensemble and the best Bayesian model, the RNN using Bayesian embeddings. In both cases, we find that all metrics are correlated with subgroup label prevalence: both uncertainty and mortality rate increase monotonically across age groups (Figure 5), and both are slightly higher in women than in men. These findings imply that random model variation during training may actually cause unintentional harm to certain patient populations, which may not be reflected in aggregate performance.

### 4.4 EMBEDDING UNCERTAINTY ANALYSIS

Another motivation for model uncertainty lies in understanding which feature values are most responsible for the variance of the predictive uncertainty distribution. Our RNN with Bayesian embeddings model is particularly well suited for this task in that the uncertainty in embedding space directly corresponds to the predictive uncertainty distribution and represents uncertainty associated with the discrete feature values. Understanding model uncertainty associated with features can allow us to recognize particularly difficult examples and understand which feature values are leading to

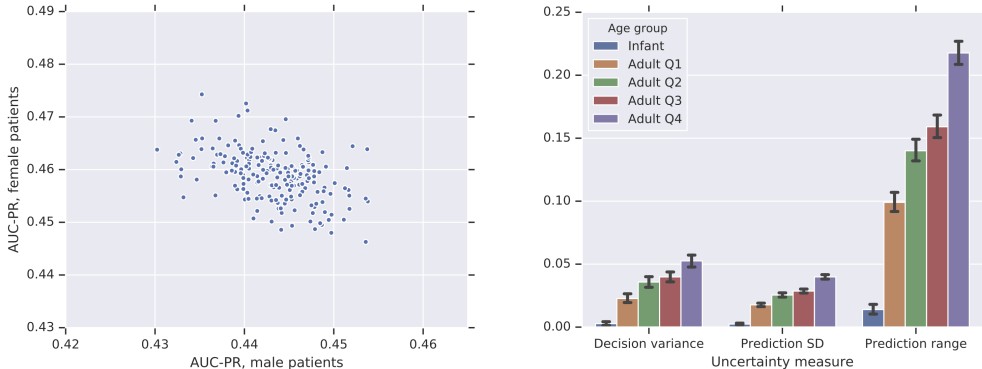

**Figure 5: Left:** Model performance comparison on male vs. female patients. Each point represents stratified AUC-PR for a single model from the deterministic ensemble. Correlation coefficient $r = -0.442$. **Right:** Summary of uncertainty measures within each age subgroup, using the Bayesian embeddings RNN. On all measures, uncertainty increases monotonically with age. This corresponds to an increase in mortality rate with age, as positive cases are more uncertain on average.

**Table 3:** Top and bottom 10 words in free-text clinical notes based on their associated Bayesian embeddings distribution's entropy, along with their frequency in the training dataset.

| | LOWEST ENTROPY | | HIGHEST ENTROPY | | |
|---|---|---|---|---|---|
| WORD | ENTROPY | FREQUENCY | WORD | ENTROPY | FREQUENCY |
| THE | -82.5445 | 41803 | 24PM | -16.0790 | 336 |
| AND | -80.6055 | 42812 | LABWORK | -16.0750 | 272 |
| OF | -80.2735 | 43191 | COLONIAL | -16.0690 | 198 |
| NO | -79.8994 | 43420 | ZOYSN | -16.0601 | 269 |
| TRACING | -78.5988 | 32181 | HT | -16.0523 | 515 |
| IS | -78.5553 | 42560 | TXCF | -15.9982 | 112 |
| TO | -77.6408 | 42365 | ARRANGEMENTS | -15.9795 | 407 |
| FOR | -76.8005 | 42972 | PARVUS | -15.9773 | 132 |
| WITH | -75.3513 | 42819 | NAS | -15.9164 | 251 |
| IN | -72.8006 | 42144 | ANESTHESIOLOGIST | -15.8796 | 220 |

the difficulties. Additionally, it provides a means of determining the types of examples that could be beneficial to add to the training dataset for future updates to the model.

For this analysis, we focus on the free-text clinical notes found in the EHR. For each word in the notes vocabulary, we have an associated embeddings distribution formulated as a multivariate Normal. We rank each word by its level of model uncertainty (measured by its embedding distribution's entropy). Table 3 lists the top and bottom 10 words, along with their frequency in the training dataset. We find that common words, both subjectively and based on prevalence counts, have low entropy and thus limited model uncertainty, while rarer words have higher levels of model uncertainty.

## 5 DISCUSSION

In this work, we demonstrated the need for capturing model uncertainty in medicine and examined methods to do so. Our experiments showed multiple findings. For example, an ensemble of deterministic RNNs captures individualized uncertainty conditioned on each patient, while the models each maintained nearly equivalent clinically-relevant dataset-level metrics. As another example, we found that models need only be uncertain around the embeddings for competitive performance, with the benefit of also enabling the ability to determine the level of model uncertainty associated with individual feature values. Furthermore, using model uncertainty methods, we examined patterns in uncertainty across patient subgroups, showing that models can exhibit higher levels of uncertainty for certain groups.

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

**Table A.1:** Hyperparameters and their associated search sets or ranges.

| HYPERPARAMETER | RANGE/SET |
|---|---|
| BATCH SIZE | {32, 64, 128, 256, 512} |
| LEARNING RATE | [0.00001, 0.1] |
| KL OR REGULARIZATION ANNEALING STEPS | [1, 1E6] |
| PRIOR STANDARD DEVIATION (BAYESIAN ONLY) | [0.135, 1.0] |
| DENSE EMBEDDING DIMENSION | {16, 32, 64, 100, 128, 256, 512} |
| EMBEDDING DIMENSION MULTIPLIER | [0.5, 1.5] |
| RNN DIMENSION | {16, 32, 64, 128, 256, 512, 1024} |
| NUMBER OF RNN LAYERS | {1, 2, 3} |
| HIDDEN AFFINE LAYER DIMENSION | {0, 16, 32, 64, 128, 256, 512} |
| BIAS UNCERTAINTY (BAYESIAN ONLY) | {TRUE, FALSE} |

**Table A.2:** Model-specific hyperparameter values.

| MODEL | BATCH SIZE | LEARN. RATE | ANNEAL. STEPS | PRIOR STD. DEV. | DENSE EMBED. DIM. | EMBED. DIM. MULT. | RNN DIM. | NUM. RNN LAYERS | HIDDEN LAYER DIM. | BIAS UNCERT. |
|---|---|---|---|---|---|---|---|---|---|---|
| DETERMINISTIC ENSEMBLE | 256 | 3.035E-4 | 1 | – | 32 | 0.858 | 1024 | 1 | 512 | – |
| BAYESIAN EMBEDDINGS | 256 | 1.238E-3 | 9.722E+5 | 0.292 | 32 | 0.858 | 1024 | 1 | 512 | FALSE |
| BAYESIAN OUTPUT | 256 | 1.647E-4 | 8.782E+5 | 0.149 | 32 | 0.858 | 1024 | 1 | 512 | FALSE |
| BAYESIAN HIDDEN+OUTPUT | 256 | 2.710E-4 | 9.912E+5 | 0.149 | 32 | 0.858 | 1024 | 1 | 512 | FALSE |
| BAYESIAN RNN+HIDDEN +OUTPUT | 512 | 1.488E-3 | 6.342E+5 | 0.252 | 32 | 1.291 | 16 | 1 | 0 | TRUE |
| FULLY BAYESIAN | 128 | 1.265E-3 | 9.983E+5 | 0.162 | 256 | 1.061 | 16 | 1 | 0 | TRUE |

# A    APPENDIX

## A.1    ADDITIONAL TRAINING DETAILS

Our RNN model uses the same embedding logic as used in Rajkomar et al. (2018a) to embed sequential and contextual features. Sequential embeddings are bagged into 1-day blocks, and fed into one or more LSTM layers. The final time-step output of the LSTM layers is concatenated with the contextual embeddings and fed into a hidden dense layer, and the output of that layer is then fed into an output dense layer yielding a single probability value. A ReLU non-linearity is used between the hidden and output dense layers, and default initializers in tf.keras.layers.* are used for all layers. More details on the training setup can be found in the code[2].

In terms of hyperparameter optimization, we searched over the hyperparameters listed in Table A.1 for the original deterministic RNN (all others in the ensemble differ only by the random seed) and each of the Bayesian models. Table A.2 lists the final hyperparameters associated with each of the models presented in the paper.

Models were implemented using TensorFlow 1.13 Abadi et al. (2016), and trained on machines equipped with Nvidia's P100 using the Adam optimizer Kingma & Ba (2014). MIMIC-III data were split into train, validation, and test set in 8:1:1 ratio.

## A.2    ADDITIONAL METRICS AND STATISTICS

Figure A.1 examines the distribution of maximum predicted probabilities over the CCS classes, along with the distribution of predicted classes associated with the maximum probabilities. Similar to the

---

[2]Code will be open-sourced.

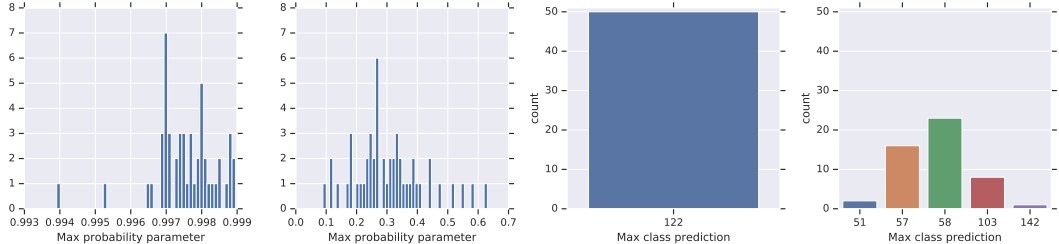

**Figure A.1: Left two:** A set of distributions for the maximum predicted probability from our deterministic ensemble for two patients in the validation dataset on the mortality task. Note the difference in x-axis scales. **Right two:** The corresponding distributions of classes associated with the max probabilities. Similar to the mortality task, for some patients, such as the one on the left, the ensemble is relatively certain about the predicted class (completely certain in this case), while for other patients, such as the one on the right, there is a larger amount of model uncertainty.

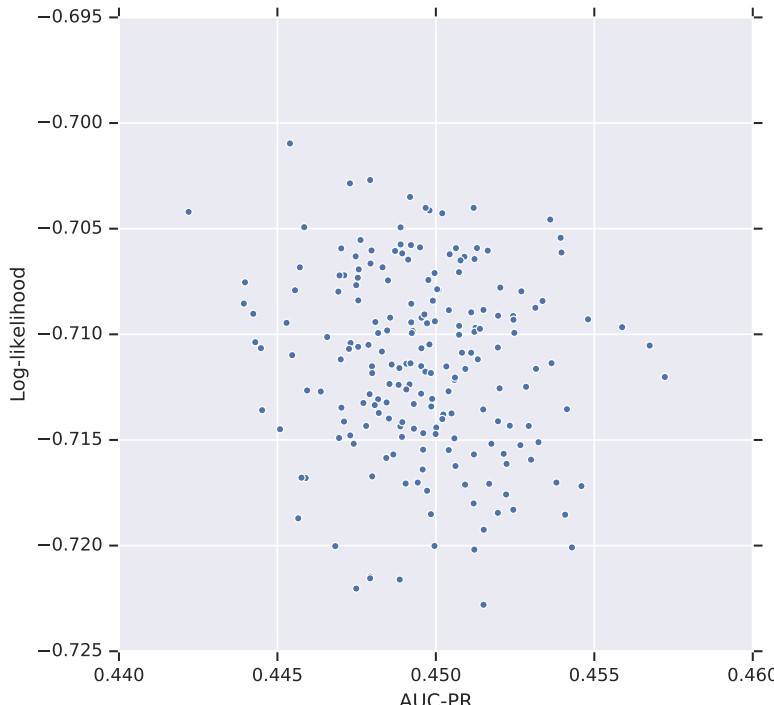

**Figure A.2:** Validation AUC-PR versus held-out log-likelihood values for the deterministic RNN ensemble on the mortality task. We find that there is no apparent correlation between the two metrics, likely due to the limited differences between the models.

binary mortality task, this demonstrates the presence of model uncertainty in the multiclass clinical setting.

In Figure A.2, we examine the correlation between held-out log-likelihood and AUC-PR values for models in the deterministic RNN ensemble on the mortality task.

In Figure A.3, we plot the differences between the maximum and minimum predicted probability values for each patient's predictive uncertainty distribution. We find that there is wide variability in predicted probabilities for some patients, and that negative patients have less variability on average.

In Table A.3, we measure the calibration of marginalized predictions of our deterministic RNN ensemble and the Bayesian RNNs.

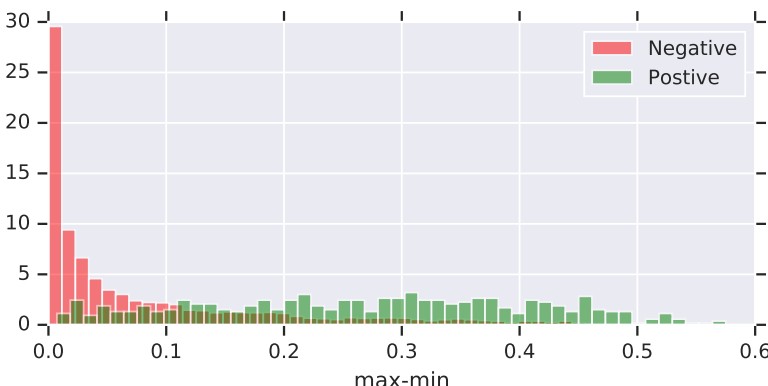

**Figure A.3:** A histogram of differences between the maximum and minimum predicted probability values for each patient's predictive uncertainty distribution. This shows that there is wide variability in predicted probabilities for some patients, and that negative patients have less variability on average.

**Table A.3:** Calibration error for marginalized predictions on the mortality task for an average over $M = 200$ models in the deterministic RNN ensemble, and $M = 200$ samples from each of the Bayesian RNN models. We find that marginalization slightly increases the calibration of the deterministic ensemble, and that the Bayesian models are comparably well-calibrated.

| MODEL | VAL. ECE | VAL. ACE | TEST ECE | TEST ACE |
|---|---|---|---|---|
| DETERMINISTIC ENSEMBLE | 0.0157 | 0.0191 | 0.0157 | 0.0191 |
| BAYESIAN EMBEDDINGS | 0.0167 | 0.0194 | 0.0163 | 0.0221 |
| BAYESIAN OUTPUT | 0.0263 | 0.0217 | 0.0241 | 0.0279 |
| BAYESIAN HIDDEN+OUTPUT | 0.0194 | 0.0212 | 0.0173 | 0.0240 |
| BAYESIAN RNN+HIDDEN+OUTPUT | 0.0240 | 0.0228 | 0.0182 | 0.0247 |
| FULLY BAYESIAN | 0.0226 | 0.0192 | 0.0178 | 0.0197 |

We additionally measure the correlation between entropy and word frequency as visualized in Figure A.4. We find further confirmation that rarer words are associated with higher model uncertainty, although there is a level of variance at a given frequency.

In Figure A.5, we plot the precision-recall (PR) curves of all $M$ models in our deterministic RNN ensemble across the full dataset along with error bars. We find that the PR curves are nearly identical for all models, and thus it again seems highly likely that any one of the models could have been selected if we were focused on our recall-based decision cost function.

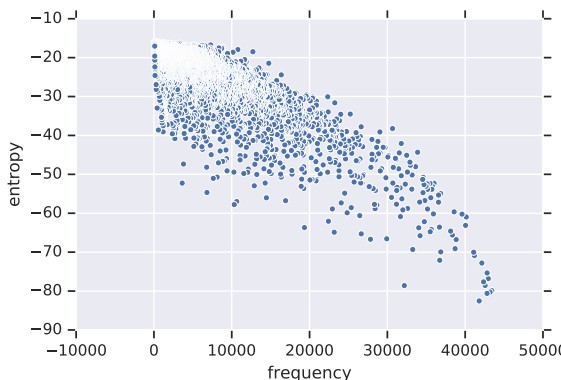

**Figure A.4:** Correlation between the entropy of the Bayesian embedding distributions for free-text clinical notes and the associated word frequency. We find that rarer words are associated with higher model uncertainty, with some level of variance at a given frequency.

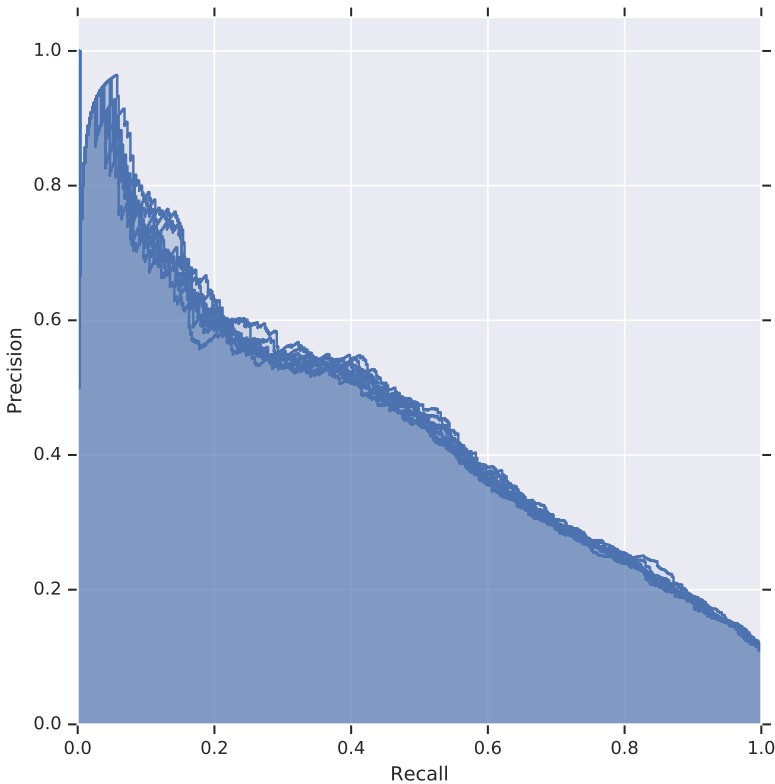

**Figure A.5:** Precision-recall curves for all $M = 200$ models in the deterministic ensemble. All of the curves are nearly identical, which is in line with the AUC-PR results.

