# OpenReview forum: "Analyzing the Role of Model Uncertainty for Electronic Health Records"
_ICLR.cc/2020/Conference — Reject_

### Official Review · AnonReviewer2 · 2019-10-22
**Official Blind Review #2**

**Rating:** 3

**Review:**

Thank you for an interesting read.

This paper is an experimental paper which argues the importance of epistemic/model uncertainty in applications for electronic health records (EHR). The main arguments are the following:
1. current metrics on dataset level cannot reveal uncertainty in prediction on personal level;
2. when evaluated on personal level, deterministic NNs with different random initialisations can produce very different predictions (thus require consideration of model uncertainty)

I am not exactly sure if ICLR is the best venue for this submission, as there is quite little innovation in modelling methodology, and the empirical analysis is domain specific. I feel this paper is more suitable to e.g. MLHC or MICCAI which focus on data analysis/machine learning methods applied to healthcare science.

Still I think the set of experiments in the paper is overall supportive to the main argument that the authors is trying to make. Possible improvements:
1. The histograms in Figure 3 & 4 clearly show that, deterministic NNs trained with different initialisations produces diverse predictions on individual patients. I commend the authors for presenting these visualisations, and I think it would be more useful to quantify this phenomenon on dataset level, e.g. compute the mean and variance of this variation of individual predictions.
2. I would expect to see an improvement of ECE for the Bayesian/deep ensemble models. The Table A.3 so marginal improvements, and I wonder how would this result support the author's claim? Also how do the ECE/ACE metrics look like when computed on sub-groups?

Apart from section 3.3, in general I think the paper writing is clear to me. The loss sensitive optimal decision method is interesting, but a lot of details are missing:
1. The presentation in section 3.3 is unclear, e.g. minimising eq. (4) w.r.t. what? What's the definition of decision region? Also what exactly is the mathematical form of the associated cost used in the experiments?
2. If I understand it correctly, in experiments the optimal thresholding method has only been applied to individual networks in the ensemble. If so what is the intention of discussing eq. (5) in the first place? Also it is unclear to me how this method performs in the deep ensemble/Bayesian RNN case. See e.g. https://arxiv.org/pdf/1805.03901.pdf for a relevant approach.

**Experience Assessment:**

I have published in this field for several years.

**Review Assessment: Checking Correctness Of Derivations And Theory:**

I assessed the sensibility of the derivations and theory.

**Review Assessment: Checking Correctness Of Experiments:**

I assessed the sensibility of the experiments.

**Review Assessment: Thoroughness In Paper Reading:**

I read the paper at least twice and used my best judgement in assessing the paper.

---

> ### Author Response · Authors · 2019-11-15
> **Response to Reviewer #2**
>
> Thank you for your feedback!
>
> > I am not exactly sure if ICLR is the best venue for this submission, as there is quite little innovation in modelling methodology, and the empirical analysis is domain specific.
>
> This paper highlights the field of medicine as an example of a field for which per-example model uncertainty is well-motivated and necessary for detecting brittle decisions.  We define brittle decisions as those for which there is high model disagreement for a given example due to model uncertainty.  This is in contrast to, say, classifying images in ImageNet, for which per-example model disagreement is currently lacking strong motivation.
>
> To date, no work has focused on the extent to which model uncertainty induces high model disagreement despite equivalent, high metric performance.  We demonstrate this novel issue and run an ablation study over various approaches to incorporating model uncertainty into recurrent models.  We show that Bayesian LSTM-based models can outperform deterministic deep LSTM ensembles, contrary to what recent literature would suggest, while still yielding qualitatively equivalent model disagreement.
>
> Thus, this work is novel and important for our field as it moves into domains with significant consequences for poor decisions.
>
> > Still I think the set of experiments in the paper is overall supportive to the main argument that the authors is trying to make.
>
> Thanks!
>
> > The Table A.3 so marginal improvements [in ECE], and I wonder how would this result support the author's claim?
>
> We show that the individual models are each well-calibrated and each have high metric performance despite the fact that they can disagree significantly for some examples.  There is limited calibration improvement when marginalized due to how well-calibrated each model was to begin with.  This strengthens the importance of the paper as it shows that it is difficult to clearly choose the best individual model or to clearly choose a marginalized prediction over the individual predictions.  Since there is high disagreement for some examples, this shows that there can be many highly-plausible models for the given data, and thus “optimal” decisions can be brittle.
>
> > minimising eq. (4) w.r.t. What?
>
> Eq. (4) is minimized with respect to the decision region $R_j$, that is, the region of the input space such that an example with input value $x$ is assigned to class $j$.
>
> > what exactly is the mathematical form of the associated cost used in the experiments?
>
> The decision loss for a specific model is a function of a probability threshold for that model, defined as
>
> $$\begin{split} L(t) = \begin{cases} \infty & \text{ if } (\operatorname{recall}(t) < \text{recall_target}) \\ -\operatorname{precision}(t) & \text{ if } (\operatorname{recall}(t) \geq \text{recall_target}) \end{cases}\end{split}$$,
>
> where $\operatorname{recall}$ and $\operatorname{precision}$ are functions that measure the recall and precision, respectively, of the model on a given dataset using a probability threshold $t$ as the class decision cutoff.  We minimize this with respect to $t$.  Practically, infinity can be substituted with a large real number.
>
> > what is the intention of discussing eq. (5) in the first place?
>
> Eq. (5) serves to represent the induced distribution over a binary optimal decision (class prediction) as a result of uncertainty in the model parameters.  In Figure 3-right, we directly show two examples: one with high agreement and one with high disagreement.  The high disagreement directly demonstrates a case where well-calibrated, high-performing individual models can disagree heavily on a given example, thus indicating a case for which the system as a whole is highly unsure about the correct decision.
>
> Interestingly, we can go a step forward.  Everything in our setup can be seen as a change of variables via functions of random variables:
>
> - Parameters $\theta \sim p_{\theta}(\theta)$
> - Predicted parameterization $\lambda \sim p_{\lambda}(\lambda | \theta, x)$ via a mapping function $f: \theta \mapsto f(\theta, x)$
> - Optimal decision $d \sim p_d(d | \lambda)$ via a mapping function $g: \lambda \mapsto g(\lambda)$
> - Utility $u \sim p_u(u | \lambda)$ via a mapping function $h: d \mapsto h(\lambda)$
>
> If $g$ and $h$ are nonlinear (just as $f$ is in our neural nets), then $\mathbb{E}[d] = \mathbb{E}[g(\lambda)] \neq g(\mathbb{E}[\lambda])$ and $\mathbb{E}[u] = \mathbb{E}[h(\lambda)] \neq h(\mathbb{E}[\lambda])$.  This shows that the data uncertainty and model uncertainty approaches are not equivalent.
>
> Furthermore, if our utility function is convex, then $h(\mathbb{E}[\lambda]) \leq \mathbb{E}[h(\lambda)]$ via Jensen's inequality.  In practice, we can approximate $\mathbb{E}[h(\lambda)]$ with a stochastic sample.  Thus, for a convex utility function, the utility using data uncertainty alone is a lower bound of the utility that could be achieved with an approach that incorporates model uncertainty.

---

### Official Review · AnonReviewer1 · 2019-10-30
**Official Blind Review #1**

**Rating:** 3

**Review:**

I am not an expert in the field of model uncertainty

Summary / contributions:
This paper discusses the important problem of model uncertainty in the output of ML models developed for clinical applications. The authors illustrate the underlying concepts using RNNs which are popular in the medical ML literature by applying these to two datasets. They argue that Bayesian RNNs with Bayesian embeddings should be the models of choice in such settings as they explicitly allow the expression of uncertainty whereby obtaining confidence intervals etc is easy. The other advantage is the fewer number of parameters that need to be stored to get such statistics.

Novelty:
-- Some of the ideas presented are standard or well-known properties to most ML practitioners. For instance, the relationship between mean and variance in Fig 2 or the uncertainty in predictions / optimal decisions in Fig 3. Is the point of the paper to make it more obvious?
-- It is certainly the case that medicine practioners are not as aware of these issues, but to reach that audience this paper would do better in a venue that caters to that community. However, the paper needs to address the concerns first so as to not confuse that community
-- The Bayesian RNN models being discussed are not novel either and their properties have been discussed in the corresponding papers (probably not in such detail and with examples).
-- What is the value of the Bernoulli distribution? Isn't the single output from a well calibrated model is enough to give the same information.

Writing:
The paper is very well written and has good figures and examples to explain the ideas. The one area that can be improved is the contributions section.


Results:
-- The authors do not discuss the related issue of model calibration in much detail. It is unclear what additional information we are gaining from the author's perspective of model uncertainty. A well calibrated model as well as other ways of obtaining confidence intervals (via hypothesis tests) would serve just as well.
-- Are the conclusions derived on the specific datasets general?
-- The results showing group-level biases are not very helpful and come across as anecdotal. These can be derived from most other models too.

**Experience Assessment:**

I do not know much about this area.

**Review Assessment: Checking Correctness Of Derivations And Theory:**

N/A

**Review Assessment: Checking Correctness Of Experiments:**

I assessed the sensibility of the experiments.

**Review Assessment: Thoroughness In Paper Reading:**

I read the paper at least twice and used my best judgement in assessing the paper.

---

> ### Author Response · Authors · 2019-11-15
> **Response to Reviewer #1**
>
> Thank you for your feedback!
>
> > Some of the ideas presented are standard or well-known properties to most ML practitioners. For instance, the relationship between mean and variance in Fig 2 or the uncertainty in predictions / optimal decisions in Fig 3.
>
> Figure 2 demonstrates standard deviation of the model disagreement over the correct predictive distribution, that is, the standard deviation of $p(\lambda | x)$.  Note that this is not the standard deviation of $p(y | x)$.  Figure 3 then shows this disagreement in both predicted probability and optimal decision spaces due to model uncertainty  This is poorly studied in research, and ML practitioners generally either use a single model or an average over a small ensemble of models.
>
> To date, no work has focused on the extent to which model uncertainty induces high model disagreement despite equivalent, high metric performance.  We demonstrate this novel issue and run an ablation study over various approaches to incorporating model uncertainty into recurrent models.  We show that Bayesian LSTM-based models can outperform deterministic deep LSTM ensembles, contrary to what recent literature would suggest, while still yielding qualitatively equivalent model disagreement.
>
> Thus, this work is novel and important for our field as it moves into domains with significant consequences for poor decisions.
>
> > What is the value of the Bernoulli distribution? Isn't the single output from a well calibrated model is enough to give the same information.
>
> We believe this is referring to Eq. (5).  This equation serves to represent the induced distribution over a binary optimal decision (class prediction in this case) as a result of uncertainty in the model parameters.  Since the decision is binary, this can be represented as a Bernoulli distribution.
>
> Interestingly, we can go a step forward.  Everything in our setup can be seen as a change of variables via functions of random variables:
>
> - Parameters $\theta \sim p_{\theta}(\theta)$
> - Predicted parameterization $\lambda \sim p_{\lambda}(\lambda | \theta, x)$ via a mapping function $f: \theta \mapsto f(\theta, x)$
> - Optimal decision $d \sim p_d(d | \lambda)$ via a mapping function $g: \lambda \mapsto g(\lambda)$
> - Utility $u \sim p_u(u | \lambda)$ via a mapping function $h: d \mapsto h(\lambda)$
>
> If $g$ and $h$ are nonlinear (just as $f$ is in our neural nets), then $\mathbb{E}[d] = \mathbb{E}[g(\lambda)] \neq g(\mathbb{E}[\lambda])$ and $\mathbb{E}[u] = \mathbb{E}[h(\lambda)] \neq h(\mathbb{E}[\lambda])$.  This shows that the data uncertainty and model uncertainty approaches are not equivalent.
>
> Furthermore, if our utility function is convex, then $h(\mathbb{E}[\lambda]) \leq \mathbb{E}[h(\lambda)]$ via Jensen's inequality.  In practice, we can approximate $\mathbb{E}[h(\lambda)]$ with a stochastic sample.  Thus, for a convex utility function, the utility using data uncertainty alone is a lower bound of the utility that could be achieved with an approach that incorporates model uncertainty.
>
> > The paper is very well written and has good figures and examples to explain the ideas.
>
> Thanks!
>
> > The authors do not discuss the related issue of model calibration in much detail. It is unclear what additional information we are gaining from the author's perspective of model uncertainty. A well calibrated model as well as other ways of obtaining confidence intervals (via hypothesis tests) would serve just as well.
>
> This is incorrect.  A single well-calibrated model cannot capture disagreement due to model uncertainty, and we show this in our paper.  In sections 2 and 4, we discuss calibration and clearly show that the individual models are each well-calibrated and each have high metric performance despite the fact that they can disagree significantly for some examples.  This strengthens the importance of the paper as it shows that it is difficult to clearly choose the best individual model.  Since there is high disagreement for some examples, this then shows that there can be many highly-plausible models for the given data, and thus “optimal” decisions can be brittle.
>
> > Are the conclusions derived on the specific datasets general?
>
> We demonstrate results on multiple datasets and multiple tasks.  Importantly, we highlight the field of medicine as an example of a field for which per-example model uncertainty is well-motivated and necessary for detecting brittle decisions.  This is in contrast to, say, classifying images in ImageNet, for which per-example model disagreement is currently lacking strong motivation.
>
> > The results showing group-level biases are not very helpful and come across as anecdotal.
>
> The group-level biases specifically demonstrate the effect of model uncertainty on different subgroups to show that the disagreement is not uniform across all subgroups.  This is incredibly important as it shows low disagreement on some sub-population will not necessarily translate to low disagreement across the population.

---

### Public Comment · ~Ethan_Steinberg1 · 2019-10-10
**No difference between model uncertainty and data uncertainty?**

One major issue with this paper is that the proposed metric, model uncertainty, is identical to data uncertainty in terms of measuring the uncertainty for an individual patient.  The fundamental problem is that a mixture of Bernoulli distributions, which the authors define as model uncertainty, contains equivalent information to a single Bernoulli distribution, which the authors define as data uncertainty. The authors don't seem to acknowledge that when a neural network outputs a single Bernoulli parameter, that parameter is already able to capture all necessary uncertainty information. (Now, whether those output Bernoulli parameters are correctly calibrated and methods for improving calibration is a completely different topic not explored in this paper).

This is a fundamental problem for the paper because the paper is structured with the objective of creating good estimates of those mixtures. Due to the equivalency mentioned above, knowing good estimates of these mixtures is not useful for understanding the uncertainty for an individual patient because the information contained in that mixture about a particular patient is identical to the information contained in a single Bernoulli centered at the mean. (Do note that knowing the mixture does provide some information about the model class, but that's a completely separate discussion and irrelevant when we are primarily concerned with knowing the uncertainty for a single prediction). The authors need to better justify why we should care about model uncertainty when data uncertainty is already able to capture all of the uncertainty in the problem.

One experiment I think would be interesting would be a thorough  comparison in AUROC and calibration between the mean Bernoulli from the RNN ensembles or Bayesian RNN compared to the single Bernoulli from a single RNN.  It's well known that ensembling is often helpful for improving calibration and accuracy, but it is often underexplored with neural networks due to limited compute resources. It would be interesting to see both the improvement on the single model due to ensembling and whether the Bayesian RNN enables you to achieve similar improvements with less compute. Note that the distribution of Bernoullis is still irrelevant for this experiment, as we can simplify that distribution into a single mean Bernoulli parameter that captures the same information and do our analysis on that parameter.

---

> ### Author Response · Authors · 2019-10-12
> **Explanation for why model uncertainty and data uncertainty are different, as well as responses to the other comments.**
>
> Thanks for taking a look at the paper.  It is indeed well-known that there exists a single Bernoulli distribution that is equivalent to a given mixture of Bernoulli distributions.  That’s always true, even for continuous distributions, and our paper doesn’t attempt to say otherwise.  However, the opposite isn't true though. That is, for a single Bernoulli distribution, there is *not* a unique mixture of Bernoulli distributions, but rather there exist infinitely many different mixtures of Bernoulli distributions.  Thus, it is a many-to-one relationship between Bernoulli mixtures and single Bernoulli distributions.  In terms of information, only the mixture of Bernoulli distributions contains the information for whether the predictive uncertainty in the final p(y | x) is due to high data uncertainty vs. high model uncertainty, and this distinction is not possible when marginalized to a single Bernoulli distribution.
>
> The distinction becomes necessary for, say, distinguishing between in-distribution and out-of-distribution examples in which there is high data uncertainty.  A single marginalized distribution does *not* carry this information.  This is shown quite clearly in section 5.1 Figure 3 of Malinin & Gales (2018):
>
>     "Figures 3b and 3c show that when classes are distinct both the entropy of the DPN’s predictive posterior and the differential entropy of the DPN have identical behaviour - low in the region of data and high elsewhere, allowing in-distribution and out-of-distribution regions to be distinguished. Figures 3e and 3f, however, show that when there is a large degree of class overlap, the entropy and differential entropy have different behavior - entropy is high both in region of class overlap and far from training data, making difficult to distinguish out-of-distribution samples and in-distribution samples at a decision boundary. In contrast, the differential entropy is low over the whole region of training data and high outside, allowing the in-distribution region to be clearly distinguished from the out-of-distribution region."
>
> Here, entropy is measured on the marginalized p(y | x), while differential entropy is measured on a distribution over p(y | x) distributions.  We already reference Malinin & Gales (2018), and would be happy to add an additional statement.  In light of the above discussion, it should be clear that our other analyses, such as the patient subgroup and input feature analyses, would not be possible with a single RNN.
>
> Regarding your comment, “Now, whether those output Bernoulli parameters are correctly calibrated and methods for improving calibration is a completely different topic not explored in this paper”, this is incorrect.  We discuss calibration in Section 2, report results for two calibration metrics (ECE and ACE) in Table 1 (with an additional table of values in A.3), and discuss them in Section 4.1, stating that the “models are overall well-calibrated”.
>
> Regarding your comments on the “comparison in AUROC and calibration” and “the improvement on the single model due to ensembling and whether the Bayesian RNN enables you to achieve similar improvements with less compute”, this information is already present in the paper in Sections 4.1 and 4.2, i.e., within our experimental results.  We state in section 4.1, “Table 1 shows the performance averaged over individual models in our deterministic ensemble, with a standard deviation in parentheses“.  Specifically (and as described in the paper), we take a deterministic RNN ensemble, measure metrics for each model within the ensemble, and report the mean and standard deviation over these individual models.  AUC-PR, AUC-ROC, ECE, and ACE are included for MIMIC-III mortality prediction.  Several metrics are also included for the multiclass CCS task as well.  In contrast, Table 2 contains marginalized metrics, i.e., metrics after averaging over the predictions from the ensemble or over samples from the Bayesian models.  Together, the two tables allow one to compare the individual model performance to marginalized model performance in the deterministic ensemble.  Furthermore, we note the computational differences in terms of total parameter counts, paired with the exact architecture and hyperparameter settings.

---

> > ### Public Comment · ~Ethan_Steinberg1 · 2019-10-12
> > **Still no justification for why data uncertainty and model uncertainty provide different information about the *uncertainty for an individual patient*?**
> >
> > Hi,
> >
> > Thanks for the thoughtful response. I totally agree that there are many possible Bernoulli mixtures for single Bernoulli, but my claim is that all of those mixtures contain identical information about the uncertainty for an individual patient. This is a problem for your paper because you primarily seem to justify model uncertainty in terms of finding uncertainty for particular patients. Let's be super explicit here with a concrete example:
> >
> > Consider the following three possible mixtures of Bernoulli parameters $\lambda$
> >
> > Mixture 1:
> > $p(\lambda = 1) = 0.5$, $p(\lambda = 0) = 0.5$, zero elsewhere.
> >
> > Mixture 2:
> > $p(\lambda = 0.7) = 0.5$, $p(\lambda = .3) = 0.5$, zero elsewhere.
> >
> > Mixture 3:
> > $p(\lambda = 0.5) = 1$, zero elsewhere.
> >
> >
> > (See https://colab.research.google.com/drive/1GWrmpA1q7XHbmxWiCLCYI7puhkhb6ACh for a histogram).
> >
> > All three of these mixtures imply that the patient outcome y follows a Bernoulli with a parameter $\lambda = .5$. Trivially, the corresponding variance of that outcome is simply $0.25$ and the corresponding entropy is -ln 2.
> >
> > Note that the entropy and variance are the same for the three mixtures. Knowing the exact mixture doesn't matter because the patient outcome follows a distribution that only depends on the mean.
> >
> > What uncertainty about the patient's outcome is not being captured in that patient outcome Bernoulli of $\lambda = .5$ (or that third "mixture")? What exact additional patient level uncertainty analysis are you saying is enabled by knowing the exact mixture? (It doesn't seem possible that something could be missed by the single Bernoulli as it fully captures the patient outcome distribution.)
> >
> > For example, your patient subgroup analysis can be performed exactly with a single RNN. Simply compute the entropy implied by the single output Bernoulli parameter.  If you don't think that's correct, or gives you suboptimal answers, you should prove it.
> >
> > You mention the possibility of distinguishing between in-distribution and out-of-distribution examples.
> > That's not directly relevant for computing the uncertainty for individual patients and if you want to make that claim, you should empirically verify that this actually helps you distinguish in-distribution to out-of-distribution compared to relevant baselines. (Alternatively, I guess you could view the in-distribution vs out-of-distribution thing as a tool for improving calibration on out of sample examples, in a sense improving your uncertainty calculations on out of sample examples. In that case your results still do simplify down to a single Bernoulli and you should compare the calibration of your method vs the calibration of the simple RNN baseline. You should also include simple baselines for improving calibration/detecting out of sample examples.)
> >
> > Similarly, your input feature analysis could not be performed, but that's simply because your input feature analysis has nothing to do with patient level uncertainty. (Or at least, you should prove that it has something to do with patient level uncertainty if you want to make that claim).

---

> > > ### Author Response · Authors · 2019-10-17
> > > **How model uncertainty provides different information for individual patients than data uncertainty.**
> > >
> > > Thank you for your detailed questions and examples. It is now much clearer that there was some confusion when talking about model uncertainty. First of all, we would like to clarify what we are proposing in this paper. We are stating that there can be multiple neural network models that show similar dataset-level metrics (i.e. AUCPR), but disagree in predictions for a certain sample. This behavior is caused by “model uncertainty”, which comes from our lack of knowledge of the exact true function, and therefore having to approximate it by training many models on a given dataset. Our claim in the paper is that we must consider model uncertainty if we are to make decisions based on neural network models (e.g. Fig 3).
> > >
> > > Given that, let us focus on your example of unanimous $p(\lambda=0.5)=1$ and dichotomous $p(\lambda=1)=0.5, p(\lambda=0)=0.5$. For a more intuitive discussion, let’s put this example in the context of doctors making a diagnosis for lung cancer. Let's describe these in the context of two patients, A and B. Former unanimous case would be all doctors diagnosing “Uncertain” for Patient A, and the latter dichotomous case would be half the doctors diagnosing “Positive”, and half the doctors diagnosing “Negative” for Patient B. If you take the mean diagnosis, it would be “Uncertain” for both patients as you pointed out.
> > >
> > > Now, we would like to point out the role of model uncertainty for individual patients in this example. Had we not considered model uncertainty, what we would do is just pick one doctor (sample one model or one $\lambda$) who has a good track record for the past several years, and accept his/her opinion (i.e. which is the same as training a single model to achieve a certain AUCPR and use that model for making predictions). Taking only this doctor’s diagnosis for Patient A (“Uncertain”) would be okay, but taking only this doctor’s diagnosis for Patient B (either “Positive” or “Negative”) could be catastrophic. This is what happens when we do not take model uncertainty into account. If we had assumed that model uncertainty could affect our decision, and asked several doctors instead of just one, we would have more insight into the status of Patient A (all doctors say “Uncertain”) and Patient B (doctors are equally divided into “Positive” and “Negative”).
> > >
> > > Your claim that a mixture of Bernoullis can be represented by a single Bernoulli, although correct, is not quite relevant to our work, because we would not be talking about a mixture of Bernoullis (i.e. many possible approximations of the true function, or metaphorically, many doctors) unless we cared about model uncertainty in the first place. Our predictions that use model uncertainty can be thought of as producing a posterior over $\lambda$.
> > >
> > > All our follow-up analyses and discussions were conducted to explain different aspects of model uncertainty. For example, to explain Figure 5 Left in the context of doctors diagnosing lung cancer; the figure shows that, even though all doctors show similar diagnosis accuracy over several years, they tend to have expertise in different subdomains. For example, some doctors might make more accurate diagnoses for male patients, while another doctor might make more accurate diagnoses for female patients. Again, if we consulted only one doctor (i.e. ignore model uncertainty), we would not always gain sufficient information. As for input feature analysis, we are analyzing which features are most responsible for creating the disagreement among doctors.
> > >
> > > As for your point about being able to do subgroup analysis with a single model; With a single model, you are indeed able to analyze the data uncertainty among different patient subgroups. But only with multiple models are you able to analyze the model uncertainty among different subgroups. Using the doctors analogy from above, model uncertainty analysis for different age groups would be saying something like "Doctors disagree more when it comes to diagnosing senior patients than children". Data uncertainty analysis for different age groups, on the other hand, would be saying something like “This doctor's predictions are more varied for senior patients than for children”. The two analyses provide us with different insights, and are not directly comparable.

---

> > > > ### Public Comment · ~Ethan_Steinberg1 · 2019-10-17
> > > > **Might to useful to carefully distinguish the difference between model uncertainty and ensembling**
> > > >
> > > > Thanks for the thorough response. Let's go back to the doctor example. It seems to me that you are mostly talking about the effectiveness of ensembling rather than simply knowing that mixture of Bernoullis.
> > > >
> > > > It seems like the crux of your argument boils down to:
> > > >
> > > > > If we had assumed that model uncertainty could affect our decision, and asked several doctors instead of just one, we would have more insight into the status of Patient A (all doctors say “Uncertain”) and Patient B (doctors are equally divided into “Positive” and “Negative”).
> > > >
> > > > What you are effectively doing in this case when you ask multiple doctors is that you are ensembling the predictions of multiple classifiers and hoping that that gives you a better result than simply taking the word of one doctor. By "insight", I assume you mean AUROC or callibration or some metric about the ability to predict patient outcomes? So to simplify your claim, it would be that: model uncertainty (the histogram of Bernoulli distributions corresponding to the model stability) is important because you can transform that histogram into a single Bernoulli (ensembling the multiple predictions) that provides better AUROC/calibration/log loss? (In other words, we should do ensembling for medical record problems because it helps quite a bit with getting better results).
> > > >
> > > > If so, then that claim can simply be answered by looking at the calibration and AUROC of the ensemble vs the single RNN. Looking at your results, the ensembling doesn't seem to add too much. Regardless, I would argue that you should reframe your paper to talk more about the effectiveness/necessity of ensembling as that's what you are actually testing.
> > > >
> > > > Going through a couple of minor ancillary points.
> > > > > Again, if we consulted only one doctor (i.e. ignore model uncertainty), we would not always gain sufficient information.
> > > > The way you can show this is by proving that the single doctor (single RNN) gives you worse calibration/AUROC/log loss etc than the ensemble.
> > > >
> > > > >As for input feature analysis, we are analyzing which features are most responsible for creating the disagreement among doctors.
> > > > Yes, but the disagreement among doctors doesn't really have much to do with patient level uncertainty. For example, consider a patient with no medical record. You are very uncertain about that patient, but you won't see that in any of your "model uncertainty" things. As before, you can simplify this to a simple calibration comparison between the ensembled vs baseline model.

---

> > > > > ### Author Response · Authors · 2019-11-15
> > > > > **Final response**
> > > > >
> > > > > The authors had a separate discussion with Ethan to discuss the concerns.  Ethan thought we were trying to show an improved optimal decision formula based on model uncertainty.  Rather, this paper is analyzing the extent of the uncertainty in predictive parameterizations and optimal decisions due to model uncertainty, and leaves model uncertainty-aware policies for improved clinical utility to future and prior work.  All concerns have been resolved, and the authors and Ethan are in agreement now.
> > > > >
> > > > > Interestingly, we can go a step forward.  Everything in our setup can be seen as a change of variables via functions of random variables:
> > > > >
> > > > > - Parameters $\theta \sim p_{\theta}(\theta)$
> > > > > - Predicted parameterization $\lambda \sim p_{\lambda}(\lambda | \theta, x)$ via a mapping function $f: \theta \mapsto f(\theta, x)$
> > > > > - Optimal decision $d \sim p_d(d | \lambda)$ via a mapping function $g: \lambda \mapsto g(\lambda)$
> > > > > - Utility $u \sim p_u(u | \lambda)$ via a mapping function $h: d \mapsto h(\lambda)$
> > > > >
> > > > > If $g$ and $h$ are nonlinear (just as $f$ is in our neural nets), then $\mathbb{E}[d] = \mathbb{E}[g(\lambda)] \neq g(\mathbb{E}[\lambda])$ and $\mathbb{E}[u] = \mathbb{E}[h(\lambda)] \neq h(\mathbb{E}[\lambda])$.  This shows that the data uncertainty and model uncertainty approaches are not equivalent.
> > > > >
> > > > > Furthermore, if our utility function is convex, then $h(\mathbb{E}[\lambda]) \leq \mathbb{E}[h(\lambda)]$ via Jensen's inequality.  In practice, we can approximate $\mathbb{E}[h(\lambda)]$ with a stochastic sample.  Thus, for a convex utility function, the utility using data uncertainty alone is a lower bound of the utility that could be achieved with an approach that incorporates model uncertainty.

---

### Decision · Program_Chairs · 2019-12-19

**Decision:**

Reject

**Comment:**

The paper considers an important problem in medical applications of deep learning, such as variability/stability of  model's predictions in face of various perturbations in the model (e.g., random seed), and evaluates different approaches to capturing model uncertainty. However, it appears to be little innovation in terms of machine-learning methodology, so ICLR might not be the best venue for this work, while perhaps other venues focused more on medical applications might be a better fit.